# Mammographic Density Changes over Time and Breast Cancer Risk: A Systematic Review and Meta-Analysis

**DOI:** 10.3390/cancers13194805

**Published:** 2021-09-25

**Authors:** Arezo Mokhtary, Andreas Karakatsanis, Antonis Valachis

**Affiliations:** 1Faculty of Medicine and Health, School of Medical Sciences, Örebro University, 70182 Örebro, Sweden; arezomokhtary99@gmail.com; 2Department of Surgical Sciences, Uppsala University, 75185 Uppsala, Sweden; andreas.karakatsanis@surgsci.uu.se; 3Department of Oncology, Faculty of Medicine and Health, Örebro University, 70182 Örebro, Sweden

**Keywords:** breast cancer risk, breast density, mammography, meta-analysis

## Abstract

**Simple Summary:**

Although mammographic density is strongly linked to the risk of breast cancer, research on the relationship between changes in density over time and the risk of breast cancer has shown conflicting results. We found in the present meta-analysis that increased breast density over time was associated with higher breast cancer risk whereas decreased breast density might be associated with lower breast cancer risk. The results of the meta-analysis constitute a potential opportunity for more individualized screening strategies based on the evolution of breast density during mammography screening.

**Abstract:**

The aim of this meta-analysis was to evaluate the association between mammographic density changes over time and the risk of breast cancer. We performed a systematic literature review based on the PubMed and ISI Web of Knowledge databases. A meta-analysis was conducted by computing extracted hazard ratios (HRs) and 95% confidence intervals (CIs) for cohort studies or odds ratios (ORs) and 95% confidence interval using inverse variance method. Of the nine studies included, five were cohort studies that used HR as a measurement type for their statistical analysis and four were case–control or cohort studies that used OR as a measurement type. Increased breast density over time in cohort studies was associated with higher breast cancer risk (HR: 1.61; 95% CI: 1.33–1.96) whereas decreased breast density over time was associated with lower breast cancer risk (HR: 0.78; 95% CI: 0.71–0.87). Similarly, increased breast density over time was associated with higher breast cancer risk in studies presented ORs (pooled OR: 1.85; 95% CI: 1.29–2.65). Our findings imply that an increase in breast density over time seems to be linked to an increased risk of breast cancer, whereas a decrease in breast density over time seems to be linked to a lower risk of breast cancer.

## 1. Introduction

Breast cancer is the most commonly diagnosed cancer and the leading cause of cancer-related death in women globally, with the majority of cases arising in Western Europe and the United States [1].

The presence of mammographically dense breast tissue, based on the amount of fibroglandular tissue in the breast representing the dense area in relation to non-dense area [2], has been shown to be a significant risk factor that contributes to breast cancer risk [3,4,5]. However, most of the studies investigating the association between mammographic density and breast cancer used only one measure of density at a single point with a significant gap between the time of the last negative mammogram and breast cancer diagnosis [5].

Mammographic density, on the other hand, is a trait that is constantly changing and normally decreases with increasing age, especially during menopause [2]. Other variables that have been proven to impact breast density are body mass index (BMI) [6], exogenous hormone use [7], diet [8,9], and reproductive history [10].

Although the breast density is expected to decrease with increased age, the breast cancer incidence increases with age, resulting in an apparent paradox on the association between breast density and breast cancer risk [11]. A proposed explanation for this paradox is that breast density mainly reflects the cumulative exposure to risk factors for breast cancer that increases the breast cancer risk [12,13].

Several studies have investigated the potential association between changes in breast density over time and breast cancer risk with conflicting results, making the interpretation of current evidence in clinical practice difficult [14,15,16,17,18,19,20,21,22].

The aim of the present systematic review and meta-analysis was to gather the current evidence and to investigate the potential association between breast density changes over time and breast cancer risk.

## 2. Materials and Methods

We searched the Pubmed and ISI Web of Science databases using the following searching algorithm:

(Breast cancer risk OR breast cancer incidence) AND (breast density OR dense breast OR dense breast tissue OR density) AND (change OR breast density change).

We considered only studies published in English to be included to our search strategy. The last search was conducted on 30th June.

The search strategy was supplemented by checking the reference lists on eligible articles. Inclusion criteria were used to screen the studies based on their title and abstract. Full-text articles were evaluated for eligibility by using the exclusion and inclusion criteria.

Two independent researchers (AM and AV) carried out the literature search, and in case of discrepancies, consensus between the researchers was achieved through discussion.

The studies that matched the following PICOS format were included in the literature review:

Population: Individuals free of breast cancer prior to study start.Intervention: Breast density changes over time.Comparison: Non-dense breast or stable breast density over time.Outcome: Invasive breast cancer (DCIS was not considered as outcome of interest).Study design: prospective or retrospective cohort studies or case–control studies.

The exclusion criteria were studies that fulfilled at least one of the following criteria: lack of data regarding density change over time; lack of data regarding breast cancer risk; no multivariate analysis investigating the association between breast density change and breast cancer risk; no adequate data to be extracted for the pooled analyses; and review articles.

Data from eligible studies were extracted by two researchers independently, and any discrepancies that arose were solved through conversation. The information that was extracted included: author; year of publication; type of study (prospective/retrospective and cohort/case-control); median follow-up; total number of patients; number of patients with breast cancer; definition of breast density change; change in breast density (non-dense stable, and decrease or increase dense stable); menopausal status; measurement type (HR = hazard ratio, RR = relative risk, and OR = odds ratio); HR/RR/OR in multivariate analysis; 95% confidence interval (CI); and covariates in multivariate analysis.

The definition of change in breast density varied among eligible studies. Four studies used BI-RADS classification (A, almost entirely fat; B, scattered fibroglandular; C, heterogeneously dense; and D, extremely dense) [14,16,17,20]. Two studies used changes in the percent of density, categorized four ways (i.e., <5%, 5–25%, >25–50%, and >50%) [19,21] One study used a fixed percent of change in the percent of density (+/−5%), using breast density changes as a continuous variable [18]. Another study used a fixed percent of changes in the relative area of mammographic density (+/−10%), using breast density changes as a continuous variable [15]. Furthermore, one study used Wolfe’s classification, with two parenchymal patterns (low-risk and high-risk patterns) [22].

The variation in definitions is a source of heterogeneity in the meta-analysis. To minimize this heterogeneity, we considered two different patterns of breast density change:Increased breast density (compared with non-dense breast) defined as A or B converted to C or D (according to BI-RADS classification); changes from <5% or 5–25% to >25–50% or >50%; or a higher percent of change than the fixed cut-off for studies using a fixed cut-off or change from a low-risk to high-risk pattern.

Considering this broad definition, one study with BI-RADS classification (or classification in four categories based on the percent of density) and separate data for changes among different categories can contribute to the meta-analysis with more than one comparison (A to C, A to D, B to C, or B to D: <5% to 25–50%, <5% to >50%, 5–25% to 25–50%, or 5–25% to >50%).

Decreased breast density (compared with stable breast density) defined as C or D converted to A or B (according to BI-RADS classification); changes from >25–50% or >50% to <5% or 2–25%; or a lower percent of change than the fixed cut-off for studies using a fixed cut-off or change from a high-risk to low-risk pattern.

As described above, one study with BI-RADS classification (or classification in four categories based on the percent of density) and separate data for changes among different categories can contribute to the meta-analysis with more than one comparison.

The Newcastle–Ottawa Quality Assessment Scale (NOS) [23] was used to assess the quality of the eligible studies. Studies having an NOS score of seven or above were judged to be of high quality. The author and supervisor completed the quality assessment individually, and any discrepancies were solved through discussion.

HR or RR, and their accompanying 95% CI from multivariate analyses on the risk for developing breast cancer depending on change in breast density over time was extracted from cohort studies, whereas OR and 95% CI were extracted from case–control studies. HR and RR were treated as similar measures of outcome for the meta-analysis whereas OR was analyzed separately.

When the same study presented data on different comparisons, we allowed more than one comparison for each study to be included to the pooled analyses in the following instances: when the different comparisons within the same study presented separate results without overlapping cohorts and when the increased or decreased breast density patterns analyzed were matched with our definitions described above.

The logHR (or RR) and standard error (SE) for each comparison was calculated, and then, the inverse variance method was used for meta-analysis of cohort studies. For case–control studies (including one cohort study presenting OR), the logOR and SE for each comparison was calculated and the inverse variance method was used for meta-analysis.

Heterogeneity between the studies was assessed using Chi^2^ test and I^2^ statistics. Indications for significant heterogeneity were *p* < 0.05 on the Chi^2^ test and I^2^ > 50%. The fixed-effects model was selected for calculating the pooled HRs (or ORs for case–control studies) in the absence of statistical heterogeneity; otherwise, the random-effects model was selected.

We also conducted meta-regression analyses to investigate the impact of study characteristics on the study estimates of relative measures. The explanatory factors in the meta-regression analyses were the sample size, the inclusion of BMI on multivariate analysis (yes vs. no), and the type of measure for breast density (continuous vs. categorical).

The results of the meta-analysis were graphically presented as forest plots and were considered statistically significant if *p* < 0.05. Publication bias was evaluated by analyzing asymmetry in SE-based funnel plots.

The meta-analysis was conducted using the Review Manager (RevMan) 5.3 software.

## 3. Results

### 3.1. Study Selection

As described in Figure 1, we found 2160 records by searching the electronic databases. The records were then screened using inclusion criteria based on their title and abstract. A total of 13 articles were selected, with 3 more being discovered by checking the references of eligible studies.

The inclusion and exclusion criteria were applied to full-text articles to find eligible studies. The following are the grounds for eliminating seven of the records: four studies did not have data regarding density change over time; two studies did not have adequate data for extraction; and one study was a review article. Thus, seven records were excluded and nine were included in the meta-analysis.

### 3.2. Characteristics of Eligible Studies

We analyzed data from nine different studies [14,15,16,17,18,19,20,21,22] published between 1998 and 2021, where six of them were designated as cohort and three were designated as case–control studies. Table 1 summarizes the characteristics of the included studies.

### 3.3. Quality Assessment of Eligible Studies

The quality of the nine eligible studies was assessed, as shown in Table 2, by using the NOS. According to the fact that studies with a NOS score of seven or above were assessed to be of high quality, seven of our eligible studies are considered high quality.

### 3.4. Increased Breast Density over Time and Breast Cancer Risk

Four cohort studies with nine comparisons (using different categories of breast density changes over time) were included in the pooled meta-analysis. The studies showed a higher risk for breast cancer in women with increased breast density compared with women with non-dense breast tissue (pooled HR: 1.61; 95% CI: 1.33–1.96, Figure 2).

From four studies that presented OR as a measure of association between increased breast density and breast cancer risk, seven comparisons were available for a pooled analysis and showed a higher risk for breast cancer with a pooled OR of 1.98 (95% CI: 1.31–3.00, Figure 3).

### 3.5. Decreased Breast Density over Time and Breast Cancer Risk

Decreased breast density over time was associated with lower breast cancer risk compared with stable breast density in 11 comparisons from 4 cohort studies (pooled HR: 0.78; and 95% CI: 0.71–0.87, Figure 4).

A similar pooled analysis on decreased breast density and breast cancer risk in four studies presenting OR (11 comparisons) showed a pooled OR of 0.90 (95% CI = 0.79–1.03) (Figure 5).

### 3.6. Meta-Regression Analyses and Publication Bias

In pooled estimates for the association between increased breast density and breast cancer risk, none of the explanatory factors were found to be statistically significant in the meta-regression analyses except from the presence of BMI in multivariate analysis that was related to lower effect size for the association (regression coefficient (β) = −0.886; *p* = 0.007).

In pooled estimates for the association between decreased breast density and breast cancer risk, none of the explanatory factors were statistically significant in the meta-regression analysis from cohort studies whereas all three factors were statistically significant in the meta-regression analysis from case–control studies (presence of BMI in multivariate analysis, β = 1.706; *p* < 0.001; sample size, β = 1.755; *p* < 0.001; type of measure for breast density, β = −2.192; *p* = 0.008).

We conducted funnel plots for each of the four meta-analyses to assess publication bias. A symmetrical distribution was demonstrated, and as a result, no indication of publications bias was found (Appendix A).

## 4. Discussion

Gathering the current evidence on changes in breast density over time and breast cancer risk, we found that breast cancer density seems to be associated to breast cancer risk in a dual manner; increased density appears to be associated with higher breast cancer risk in both cohort and case–control studies, whereas decreased breast density appears to be associated with a reduced risk of breast cancer in cohort studies.

The findings from the present meta-analysis support a recent biological hypothesis on the association between breast density and breast cancer risk. Boyd et al. incorporated the Moolgavkar model of carcinogenesis into radiological findings of breast tissue from healthy subjects and found that dense breast tissue might promote the accumulation of mutations responsible for transition to malignant cells and that the cumulative exposure to dense breast tissue seems to be associated with the age-incidence curve of breast cancer [24]. In other words, according to the present meta-analysis, patients with a longer period of high breast density, such as those with increased breast density over time, might be at higher risk for accumulating genetic alterations of significance for carcinogenesis, whereas patients with decreased breast density over time have a shorter period of high breast density and eventually lower accumulation of genetic alterations. Such genetic alterations might be seen in not only the nuclear genome but also in the mitochondrial genome, where DNA mutations and epigenetic modifications also have been associated with breast cancer development [25].

Several challenges related to the design of eligible studies needed to be addressed during the meta-analysis. First, both cohort studies presenting HR or RR, and case–control studies presenting OR were available. We decided to perform separate meta-analyses for the different study designs since case–control studies are often considered lower in the hierarchy of evidence compared with cohort studies. Although the pooled estimates from both cohort and case–control studies have the same direction, namely towards a positive association between increased breast density and breast cancer risk and a negative association between decreased breast density and breast cancer risk, the latter was more evident in pooled analyses of cohort studies than of case–control studies. However, the pooled analysis of case–control studies might be influenced by the fact that one comparison weighted for nearly 75% of included studies. This observation, along with the higher risk of bias with case–control studies in general, led us to conclude that the pooled analyses of cohort studies can be considered more reliable.

Another challenge with this meta-analysis was the different definitions of breast density change that were used in eligible studies included in the meta-analysis. In fact, there were studies using quantitative criteria (based on an analysis of the breast density changes as a continuous variable) and others using qualitative criteria (BI-RADS classification or percent breast density in categories). To limit the risk of heterogeneity due to different definitions, we considered two scenarios. The first scenario included all comparisons among and within studies where a clear definition of increased breast density over time could be extracted. The second scenario included all comparisons among and within studies where a clear definition of decreased breast density over time could be identified. These two broad scenarios enabled us to include several comparisons in pooled estimates and to offer a more general aspect on the potential impact of changes in breast cancer on breast cancer risk. However, this approach can influence the interpretation of results regarding which classification system to use.

Looking at study-level results, the use of qualitative classifications systems for breast density seems to be more often associated with breast cancer risk, compared with a quantitative analysis of breast density [14,15,16,17,18,19,20,21,22]. However, the type of measure for breast density was not found to be statistically significant as an explanatory factor in any of the meta-regression analysis performed. Given the substantial interobserver agreement with BI-RADS classification [26], especially when the fifth version is used [27]; the evidence from individual studies [14,16,17,20]; and the current meta-analysis, and the fact that qualitative classification systems are more easily implemented in clinical practice, BI-RADS classification seems to be the most appropriate classification system for breast cancer density changes in association with breast cancer risk.

An additional challenge with this meta-analysis was the consideration of confounding factors influencing both breast density and breast cancer risk such as BMI, reproductive factors, age, and exogenous hormones. To overcome this potential source of bias, we included in our pooled estimates only comparisons using multivariate analyses considering potential confounders. However, several studies [17,18,19,20,21,22] were unable to adjust for important confounders due to a lack of adequate data, which can potentially influence the results. Given the importance of BMI in breast density, we used the presence of BMI in the multivariate analysis as an explanatory factor in our meta-regression analyses and confirmed the significance of this factor as it was associated with a lower effect size in the meta-analysis of case–control studies. As a result, BMI should always be considered as a confounding factor when the association between breast density and breast cancer risk is investigated, and appropriate statistical measures should be applied to deal with this confounder.

Two studies [13,28] investigating the association between breast density changes and breast cancer risk could not be included in the current meta-analysis due to a lack of adequate data for meta-analysis. Both were case–control studies that analyzed breast density as a continuous variable and found no statistically significant association between changes in breast density and breast cancer risk. Although the inclusion of these two studies in the pooled estimates might influence the magnitude of effect observed in our meta-analysis, the pooled estimates from cohort studies would remain unchanged. Interestingly, it has been argued that the changes in breast density should be sufficiently large to be associated with a meaningful higher or lower breast cancer risk and that this sufficient large change is observed in quantitative classification but not in qualitative ones [16,17].

A clinically relevant aspect that should be discussed is the fact that all pooled estimates referred to breast cancer patients in general and not to specific subgroups of potential interest. Due to a lack of adequate data, we were unable to perform subgroup analyses based on menopausal status or on family history of breast cancer. Considering the expected changes in breast density between pre- and postmenopausal women, separate analyses would be of interest and should be included in future studies on this topic. In addition, prior studies found that the association between breast density and breast cancer risk was stronger in women with a family history of breast cancer [29], suggesting that this patient group should be further explored.

## 5. Conclusions

In summary, we found a statistically significant and clinically relevant association between changes in breast density and breast cancer risk, mainly in terms of higher risk with increased breast density, where the pooled estimates were consistent in both cohort and case–control studies. Despite the methodological challenges, the results imply that breast density changes over time might be an opportunity for a more individualized screening strategy. Further studies are needed to evaluate whether escalated screening strategies with ultrasound [30], tomosynthesis or breast-MRI [31] would result in higher breast cancer detection rates in patients with increased breast density over time.

## Figures and Tables

**Figure 1 cancers-13-04805-f001:**
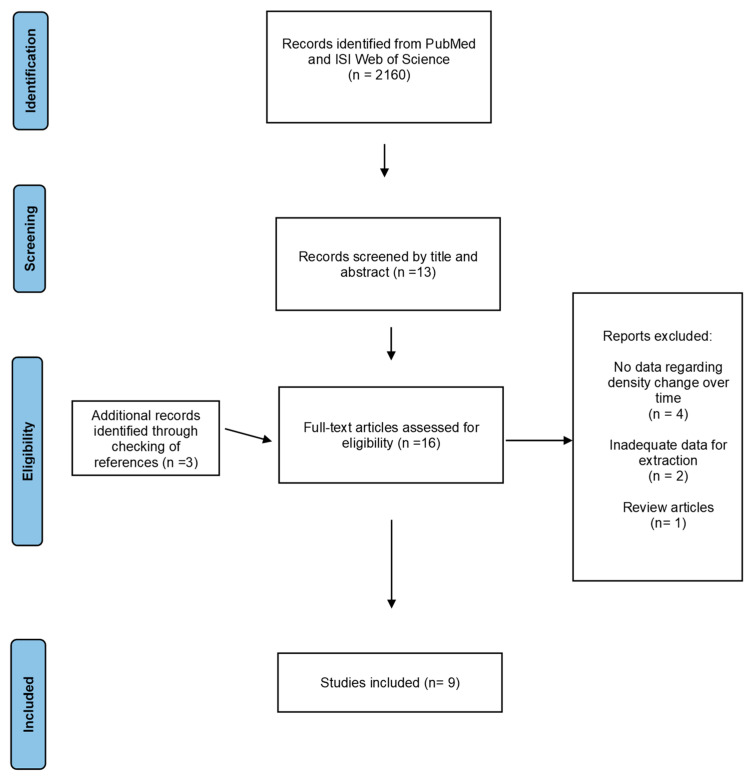
Flowchart of the study selection process.

**Figure 2 cancers-13-04805-f002:**
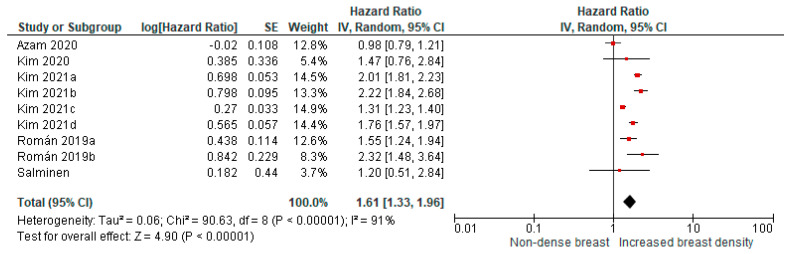
Pooled analysis for breast cancer risk in relation to increased breast density over time (cohort studies). Kim 2021a: A to C; Kim 2021b: A to D; Kim 2021c: B to C; Kim 2021d: B to D; Román 2019a: B to C; and Román 2019b: B to D.

**Figure 3 cancers-13-04805-f003:**
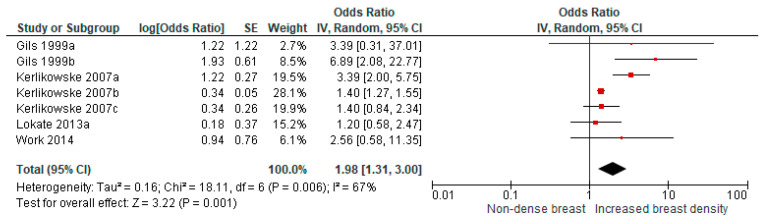
Pooled analysis for breast cancer risk in relation to increased breast density over time (studies presenting odds ratios). Gils 1999a: <5% to >25%; Gils 1999b: 5–25% to >25%; Kerlikowske 2007a: A to C; Kerlikowske 2007b: B to C; and Kerlikowske 2007c: B to D.

**Figure 4 cancers-13-04805-f004:**
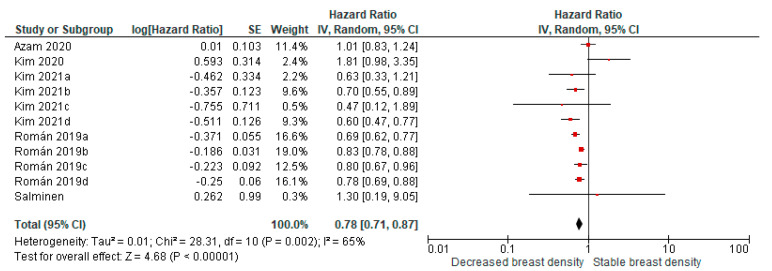
Pooled analysis for breast cancer risk in relation to decreased breast density over time (cohort studies). Kim 2021a: C to A; Kim 2021b: C to B; Kim 2021c: D to A; Kim 2021d: D to B; Román 2019a: C to A; Román 2019b: C to B; Román 2019c: D to A; and Román 2019d: D to B.

**Figure 5 cancers-13-04805-f005:**
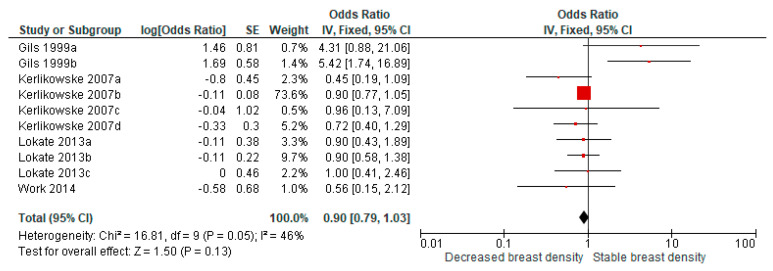
Pooled analysis for breast cancer risk in relation to decreased breast density over time (studies presenting odds ratios). Gils 1999a: >25% to <5%; Gils 1999b: >25% to 5–25%; Kerlikowske 2007a: C to A; Kerlikowske 2007b: C to B; Kerlikowske 2007c: D to A; Kerlikowske 2007d: D to B; Lokate 2013a: 25–50% to <5%; Lokate 2013b: 25–50% to 5–25%; and Lokate 2013c: >50% to 5–25%.

**Table 1 cancers-13-04805-t001:** Characteristics of eligible studies.

First Author [Ref]	Year of Publication	Type of Study	Number of Patients	Median Follow-Up	Definition of Breast Density Change
S. Kim [14]	2021	Retrospective cohort	3,301,279	NR	Dense breast = heterogeneously (C) or extremely (D) according to BI-RADS
Azam [15]	2020	Prospective cohort	43,810	5.4 years	Relative area MDC >10% increased <10% decreased
E. Kim [16]	2020	Prospective cohort	74,249	6.1 years	Dense breast = heterogeneously (C) or extremely (D) according to BI-RADS
Roman [17]	2019	Retrospective cohort	117,388	4.1 years	Dense breast = heterogeneously (C) or extremely (D) according to BI-RADS
Work [18]	2014	Prospective case–control	170	4 years	Percent density change <−5% decrease >5% increase −5% to 5% stable
Lokate [19]	2013	Retrospective case–control	1900	10 years, cases 11 years, control	Percent density at the initial and last mammogram <5% 5–25% >25–50% >50%
Kerlikowske [20]	2007	Prospective cohort	301,955	NR	Dense breast = heterogeneously (C) or extremely (D) according to BI-RADS
Gils [21]	1999	Prospective case–control	400	NR	Percent density at the initial and last mammogram <5% 5–25% >25%
Salminen [22]	1998	Retrospective cohort	4163	NR	According to Wolfe’s classification N1, P1 = Low-risk pattern P2, DY = High-risk pattern

Abbreviations: Ref, reference; NR, not reported, MDC, mammographic density change.

**Table 2 cancers-13-04805-t002:** Quality assessment of eligible studies. Cohorts and case–control studies presented separately.

	Selection				Comparability		Outcome			
NOS Questions	1	2	3	4	5	6	7	8	9	Total Score
Cohort studies										
Kim [14]	*	*	*	*	*	*	*	0	0	7/9
Azam [15]	*	*	*	*	*	*	*	0	0	7/9
Kim [16]	*	*	*	*	*	*	*	*	0	8/9
Roman [17]	*	*	*	*	*	0	*	0	0	6/9
Kerlikowske [20]	*	*	*	*	*	*	*	0	0	7/9
Salminen [22]	*	*	*	*	0	0	*	0	0	5/9
Case–control studies										
Work [18]	*	*	0	*	*	*	*	*	*	8/9
Lokate [19]	*	*	*	*	*	*	*	*	*	9/9
Gils [21]	*	*	*	*	0	0	*	*	*	8/9

Abbreviations: NOS, Newcastle–Ottawa Scale. * A star represents that the study fulfilled the quality criterion for this perspective.

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
