# Peer review of "Mammographic Density Changes over Time and Breast Cancer Risk: A Systematic Review and Meta-Analysis"

_cancers, 2021, doi:10.3390/cancers13194805_

Round 1

Reviewer 1 Report

Minor comments are as follows:

Materials and Methods

  • Page 2, Lines 63 and 77: "we considered only studies published in English to be included to our 77 searching strategy" is stated twice, please remove one of them.
  • Page 2, Line 75:  "Outcome: Breast cancer" - does this include both invasive carcinoma and ductal carcinoma in situ? Please clarify.

Reviewer 2 Report

In the manuscript by  Mokhtary et al. the authors summarize recent data and research achievements on the relation between breast density and breast cancer risk. The review is comprehensive and provides a potential risk factor for breast cancer, which has guiding significance for subsequent research and treatment. I truly enjoyed reading the manuscript and would like to give a few suggestions.

1.Questions about comparisons selection

1.1.Page 6, line 170-171:"Four cohort studies with nine comparisons (using different categories of breast density changes over time)"Could authors clarify the criteria for selecting the comparisons?

2.Some discussions and citations are missing

2.1.Page 8, line 223-232:" Boyd et al. incorporated 223
the Moolgavkar model of carcinogenesis into radiological findings of breast tissue from healthy subjects and found that dense breast tissue might promote the accumulation of mutations responsible for transition to malignant cells and that the cumulative exposure to dense breast tissue seems to be associated with the age-incidence curve of breast cancer. In other words, patients with a longer period of high breast density, as those with increased breast density over time according to the present meta-analysis, might be at higher risk for accumulating genetic alterations of significance for carcinogenesis whereas patients with decreased breast density over time have a shorter period of high breast density and eventually lower accumulation of genetic alterations."

There is another review also showed the importance of mitochondrial gene mutation leading to breast cancer development, which might be also related to breast density.(Please cite Chen et al. Semin Cancer Biol. 2020 Oct 6 doi:10.1016/j.semcancer.2020.09.012.)

2.2.Can authors discuss more about the underlying causes of breast density? Thus, it can not only reveal to other researchers that breast density is a direct risk factor for breast cancer, but also point out the future research direction for researchers to find the fundamental risk factors of breast cancer and discover the potential mechanisms.

Best regards

Round 2

Reviewer 2 Report

Authors made correction according to my previous suggestions. Strongly recommend for publication.

Best

Author Response

We thank the reviewer for the recommendation. We have checked the manuscript regarding spelling errors and edited accordingly.